RBI-ThPhys-2021-19

# Odd thermodynamic limit for the Loschmidt echo

Gianpaolo Torre,[1] Vanja Marić,[2, 3] Domagoj Kuić,[2] Fabio Franchini,[2] and Salvatore Marco Giampaolo[2]

[1]*Department of Physics, Faculty of Science, University of Zagreb, Bijenička cesta 32, 10000 Zagreb, Croatia.*
[2]*Ruđer Bošković Institute, Bijenička cesta 54, 10000 Zagreb, Croatia*
[3]*SISSA and INFN, via Bonomea 265, 34136 Trieste, Italy*
(Dated: August 16, 2022)

Is it possible to readily distinguish a system made by an Avogadro's number of identical elements and one with a single additional one? Usually, the answer to this question is negative but, in this work, we show that in antiferromagnetic quantum spin rings a simple out-of-equilibrium experiment can do so, yielding two qualitatively and quantitatively different outcomes depending on whether the system includes an even or an odd number of elements. We consider a local quantum-quench setup and calculate a generating function of the work done, namely, the Loschmidt echo, showing that it displays different features depending on the presence or absence of topological frustration, which is triggered by the even/oddness in the number of the chain sites. We employ the prototypical quantum Ising chain to illustrate this phenomenology, which we argue being generic for antiferromagnetic spin chains, as it stems primarily from the different low energy spectra of frustrated and non frustrated chains. Our results thus prove that these well-known spectral differences lead indeed to distinct observable characteristics and open the way to harvest them in quantum thermodynamics protocols.

## 1. INTRODUCTION

Quantum dynamics has been a very active field of research in the new century since sufficiently weak system-environment couplings have been achieved with ultra-cold atoms on optical lattices [1–3], enabling us to perform reliable experiments on the unitary dynamics of closed quantum systems. Stimulated by the experimental progress, theoretical questions about relaxation and the presence or absence of thermalization [4–7] have been studied intensively. Perhaps the most widely studied, and simplest, way of bringing a system out of equilibrium is the *quantum quench* protocol [5, 8–10]. Here, the system is prepared in the ground state of an initial Hamiltonian and it is suddenly let to evolve unitarily by a different Hamiltonian, obtained, for example, by changing one of the system parameters.

Developments in quantum dynamics have led to conceptual advancements in the foundations of statistical mechanics [11–14]. Furthermore, even well-established concepts, such as quantum phase transitions [15], have received their characterizations in terms of dynamic quantities. Quantum phase transitions (QPTs) are points of non-analyticity of the ground state energy, that are accompanied by gapless energy spectra and changes in the macroscopical behavior of systems [15]. In a dynamical setting, there is, for instance, evidence [16] that order parameters are best enhanced for quenches in the vicinity of quantum critical points. However, even a more basic quantity, the Loschmidt echo (LE) [17, 18] has been proposed to be used as a witness of quantum criticality [19].

This last quantity, i.e., the Loschmidt echo $\mathcal{L}(t)$, is defined as the squared absolute value of the overlap between the initial and the evolved state at time $t$. If the system is prepared in the ground state $|g\rangle$ of the initial Hamiltonian $H_0$, and then, suddenly, at $t = 0$, an unitary evolution is induced by the Hamiltonian $H_1$, then the LE can be defined as

$$\mathcal{L}(t) = |\langle g|e^{-\imath H_1 t}|g\rangle|^2. \tag{1}$$

Following the initial work [19], substantial evidence [20–28] has been collected that the LE of quenches to quantum criticality is characterized by an enhanced decay and periodic revivals, although there are known exceptions [29]. Importantly, LE can be experimentally measured, for instance by coupling the system of interest to an auxiliary two-level system, where the LE is the measure of the decoherence of the auxiliary system [17, 19, 22, 30]. It has been also stressed that the LE is readily related to the work probability distribution [31] and thus characterizes the performance of quantum systems with various thermodynamic protocols [32].

In this work, we show that the LE can be used to distinguish an antiferromagnetic spin-1/2 ring consisting of $N$ elements (sites) from the one consisting of a single additional element, i.e. of total $N+1$ elements, for arbitrarily large $N$. It is known that in such systems, depending on whether we follow the even or odd system sizes towards the thermodynamic limit, the energy spectrum is gapped or gapless respectively [33], due to the presence of topological frustration for odd $N$. Thus, similarly to bringing the system to criticality, topological frustration closes its spectral gap, although the gapless excitations in frustrated chains are not relativistic. While the spectral differences for even and odd $N$ have been traditionally deemed inconsequential, we exploit them to construct a (local) quantum quench protocol in which the LE displays different features, qualitatively and quantitatively, for the two cases. Consequently, measuring the LE in an experiment enables distinguishing systems made of $N$ and $N + 1$ spins, for arbitrarily large $N$.

Indeed, this is not the first time that the even/oddness in the number of elements of a system turns out to have physical consequences. For instance, in Ref. [34] a dependence in the current/voltage curve of a superconducting

transistor on the parity of the number of electrons in the dot was observed and discussed. We remark, however, that the latter is a mesoscopic effect, while the phenomenon we discuss persists in systems several orders of magnitude bigger.

## 2. GENERAL SCHEME

Our result follows a series of works that have shown striking differences in spin chains with an even or odd number of sites in a static setting, in a seeming violation of the usual assumption that boundary conditions are irrelevant in the thermodynamic limit for systems with a finite correlation length (for critical systems the role of boundary conditions has been understood long ago, see, for instance, Refs. [35, 36]). For instance, it was shown that the usual finite order parameter that exists for even $N$ can turn to zero for odd $N$ [37] or acquire an incommensurate modulation over the chain [38], with a first-order boundary phase transition separating the two behaviors. Moreover, as the order parameter can vanish on both sides of a phase transition, and be substituted by a string order, the nature of a second-order QPT can change with the addition of a single site [39]. While these results are recent and related to a static setting, they spring from the established spectral difference of frustrated systems which directly lends itself to a dynamical analysis.

In fact, a deeper insight in the time behavior of the LE can be obtained by expanding the initial state in terms of the eigenstates $|n\rangle$ of the perturbed Hamiltonian $H_1$:

$$\mathcal{L}(t) = \left| \sum_n e^{-iE_n t} |c_n|^2 \right|^2, \qquad c_n = \langle n|g \rangle. \qquad (2)$$

In the general (nontrivial) case, the state $|g\rangle$ is not an eigenstate of the Hamiltonian $H_1$ and thus several coefficients $c_n$ assume a non-vanishing value, and the time evolution of the LE depends on their relative weights. Roughly speaking, we can arrange the possible behaviors into two large families. The first is made of the cases in which one of the coefficients is much greater, in absolute value than the sum of all the others. Denoting by $|0\rangle$ the eigenstate of $H_1$ for which $c_n$ reaches the maximum, from eq. (2) we recover that the LE will be characterized by oscillations with an average value close to the identity and oscillation amplitudes bounded from above by $(1 - |c_0|^2)|c_0|^2$. On the other hand, if none of the $c_n$ dominates over the others, we can obtain an evolution characterized by a more complex pattern with larger oscillation amplitudes.

These two prototypical behaviors for the LE are generally associated with different properties of the physical systems [40, 41]. For example, the first trend type characterizes systems in which $H_0$ shows an energy gap that separates the ground state from the set of the excited states [21, 22, 31]. Let us consider a quench with

$H_1 = H_0 + \lambda H_p$, where $\lambda$ is the parameter whose non-zero value brings the system out of equilibrium and the eigenvalues of $H_p$ are of the order of unity [18, 42, 43]. In this case, assuming that $\lambda$ is much smaller than the energy gap, the coefficient $\langle g_1|g \rangle$, where $|g_1\rangle$ is the ground state of $H_1$, is expected to be much larger than all the others. Consequently, LE is expected to display the dynamics of the first kind. On the other side, for systems in which the ground state of $H_0$ is a part of a narrow band that, in the thermodynamic limit, tends to a continuous spectrum, as at quantum criticality, the perturbation $\lambda H_p$ may induce a non-negligible population in several low-energy excited states [44], resulting in the time evolution of the second kind.

Typically, these different spectrum properties do not turn into one another by changing the number of elements that make up the system. Indeed, the presence or the absence of the gap in the energy spectrum is related to the different symmetries of the Hamiltonian, which are, usually, size independent. Hence, keeping all other parameters fixed and increasing the number of elements, we expect the same kind of time evolution, with finite-size effects that reduce with the system size up to some point at which the dependence of the LE on the number of constituents is almost undetectable. To have a LE evolution that changes as the number of elements turns from even to odd and vice versa, we need a system in which also the shape of the energy spectrum is strongly dependent on it.

Such models can be found among the one-dimensional spin-1/2 models with topological frustration [45–51]. Namely, these are short-range antiferromagnetic systems with periodic boundary conditions in which frustration is induced when the number of elements making up the system is odd, so realizing the so-called frustrated boundary conditions [33, 37, 52, 53]. Hence the presence/absence of frustration in the ring geometry (therefore also the term *topological frustration*) is a direct consequence of the fact that the number of spins is odd/even. This particular dependence of the spectrum on the length of the ring is not found in all frustrated models. For example, models with a higher degree of frustration, such as the one-dimensional spin-1/2 quantum ANNNI model [54–56], display a phase in which the gap closes as a function of the number of spins regardless of whether this number is even or odd.

To better understand how the topological frustration works, let us take a step back. In classical antiferromagnetic systems, when the number of the elements is an integer multiple of two, even in the presence of periodic boundary conditions, there are no problems in minimizing the contribution of every single term to the total energy. Therefore, the system will show a ground state manifold made of the two Néel states, well separated from the excited states by an energy gap that does not vanish in the thermodynamic limit. This picture is very resilient and also the introduction of quantum effects does not change it significantly [57, 58]. On the contrary,

when the number of elements is odd, the presence of the periodic boundary conditions makes it impossible to satisfy simultaneously all the local constraints [45, 46]. This impossibility induces frustration, which gives rise to the creation of a set of states that are Néel states with a pair of parallelly oriented spins, the so-called domain wall. If the system is invariant under spatial translation since the defect can be placed equivalently on every lattice site, the ground state manifold of the system becomes highly degenerate, consisting of $2N$ states for a chain made of $N$ sites, and it is separated from the other states by an energy gap that stays finite in the thermodynamic limit. When quantum effects are taken into account, the macroscopic ground-state degeneracy is typically lifted, generating a narrow band of states (which can be interpreted as containing a single excitation with a definite momentum) and thus yielding an energy gap that vanishes in the thermodynamic limit. Also this picture is extremely general and characterizes both integrable and non-integrable models [37]. Hence, as a result, the energy spectrum of such models depends dramatically on whether the number of elements is even or odd.

However, this property alone is not sufficient to ensure a dependence of the dynamics of the LE on the size of the system like the one we are looking for. The perturbation that acts on the initial Hamiltonian must also be chosen carefully. On the one hand, as the states in the lowest energy band of the frustrated system are identified by different quantum numbers (namely, their momenta), the perturbation should break the symmetry these numbers reflect, to ensure that the eigenstates of the perturbed Hamiltonian can have a finite overlap in the whole band (otherwise, the initial state would overlap only with states carrying the same quantum number). On the other hand, if the unfrustrated system is in a symmetry-broken phase with an (asymptotically) degenerate ground-state manifold, we want the perturbation to preserve the ground state choice so that in the evolution the overlap between other ground state vectors remain suppressed.

## 3.   A PARADIGMATIC EXAMPLE

To clarify these arguments and to provide a specific example, let us discuss a paradigmatic model, i.e. the antiferromagnetic Ising chain in a transverse magnetic field with periodic boundary conditions [15, 59, 60].

$$H_0 = \sum_{j=1}^{N} \left( \sigma_j^x \sigma_{j+1}^x + h \sigma_j^z \right). \tag{3}$$

This model, like other more complex ones, shows the properties of the spectrum we have just identified [52] but, thanks to its simplicity, it admits an analytical solution based on a mapping to a model of free fermions. In (3) $\sigma_j^\alpha$ with $\alpha = x, y, z$ stands for the Pauli operators defined on the $j$-th lattice site, $h$ is the relative weight of

the local transverse field, $N$ is the length of the ring and periodic boundary conditions imply that $\sigma_{N+j}^\alpha = \sigma_j^\alpha$. As we can see from eq. (3), the system has the parity symmetry with respect to the $z$-spin direction, i.e. $[H_0, \Pi^z] = 0$, where $\Pi^z = \bigotimes_{i=1}^{N} \sigma_i^z$. This means that the eigenstates of $H_0$ can be arranged in two sectors, corresponding to two different eigenvalues of $\Pi^z$. Moreover, the model in eq. (3) is also invariant under spatial translation which implies that there exists a complete set of eigenstates of $H_0$ made of states with definite lattice momentum [38].

In the range $0 < h < 1$, for $N = 2M$ the system, shows two nearly degenerate lowest energy states with opposite parity and an energy difference closing exponentially with the system size [60, 61] while all the other states remain separated by a finite energy gap. When $N = 2M+1$, topological frustration sets in, and the unique ground state is part of a band in which states of different parities alternate. In this case, the gaps between the lowest energy states close algebraically as $1/N^2$ [52, 53, 62–66].

A simple perturbation that satisfies the criteria we discussed above is $H_p = \sigma_N^z$, breaking the translational invariance which classifies the eigenstates of $H_0$ while preserving the parity symmetry. Thus, we have

$$H_1 = H_0 + \lambda \, \sigma_N^z, \tag{4}$$

and we assume that the amplitude $\lambda$ is much smaller than the energy gap above the two quasidegenerate ground states in the unfrustrated case, i.e. $\lambda \ll 1$, and size-independent. Obviously other quenches, including global ones, where $H_1$ preserves the parity and breaks the translational invariance, for example through a modulated or a random transverse field, could also be considered. However, the perturbation must not be large enough to complicate the evolution of the LE in the unfrustrated case, which we want to be of the first kind. Furthermore, we are looking for a simple, unambiguous, system-size independent protocol, for which the local quench is suitable.

Since $H_1$ is not invariant under spatial translations, we cannot diagonalize it analytically as is possible for $H_0$, by exploiting the usual approach based on the Jordan-Wigner transformation followed by a Bogoliuov rotation [60]. Nevertheless, we can resort to the diagonalization procedure reported in [57], which allows us to diagonalize numerically the Hamiltonians (3) and (4) in an efficient way [67], and thus to calculate the LE (see the Methods section for the details). The results obtained are summarized in Fig. 1, where several behaviors of the LE for even $N$ and odd $N + 1$ sizes are compared.

The results fit well in the qualitative picture we have discussed in the first part. When $N$ is even and hence the system is not frustrated the LE presents small noisy oscillations around a value close to unity, see Fig. 1. The average value is almost independent of the parameters, while oscillations reduce as the system size increases. This behavior reflects the fact that $\lambda$ being small, the initial state shares a significant overlap only with one of the lowest eigenstates of $H_1$, and the contributions from all other

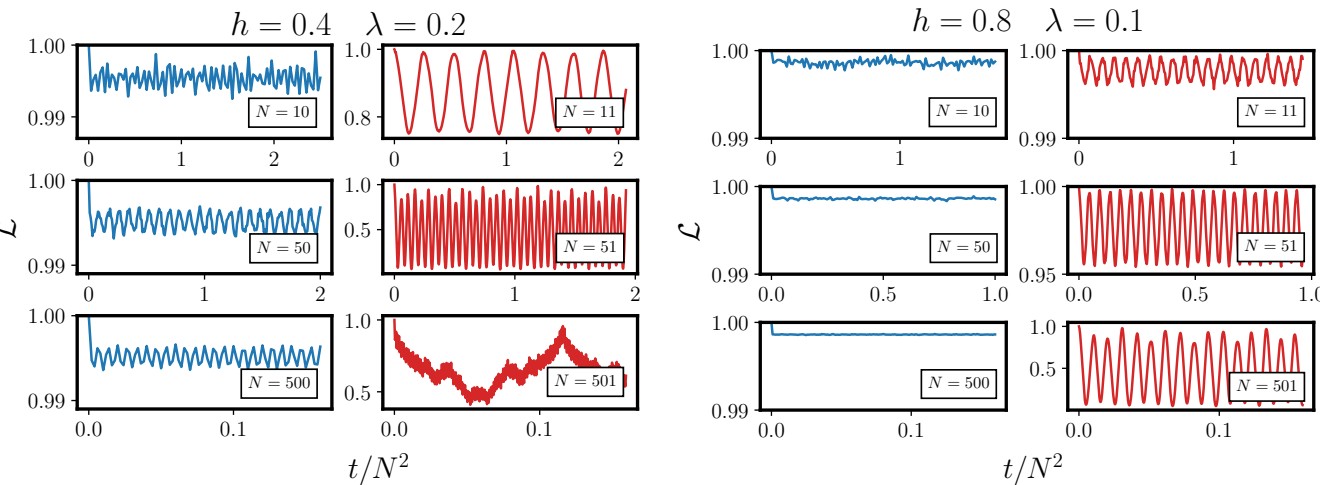

FIG. 1: (Color online) Loschmidt echo comparison between frustrated and unfrustrated chains of similar length $N$, fixing the magnetic field and the perturbation parameter respectively to $h = 0.4$, $\lambda = 0.2$ (left plot) and $h = 0.8$, $\lambda = 0.1$ (right plot). The time is rescaled for a better comparison. For even $N$ (unfrustrated systems), due to the negligible hybridization with the first excited states, the LE presents small oscillations around a value near one (left columns). For odd $N$ instead the higher number of hybridized states results in a strong sensitivity of the LE oscillations to the system parameters.

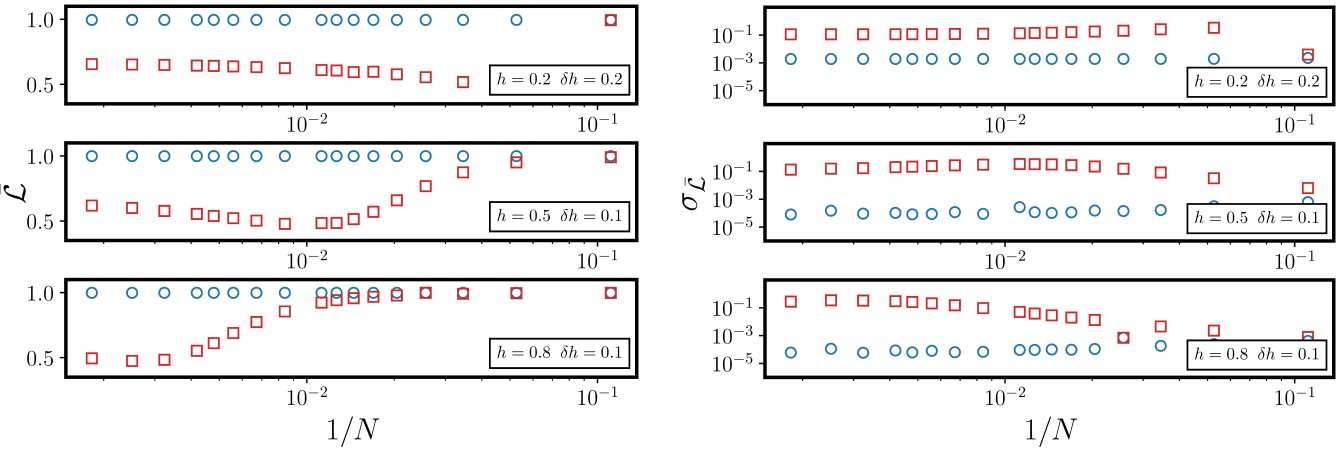

FIG. 2: (Color online) Comparison between the result for frustrated (red squares) and unfrustrated (blue circles) chains of the time-average (left panel) and of the standard deviations (right panel) of the LE for several sets of parameters as a function of the inverse system length. Differently from the frustrated case, the unfrustrated time average is mostly size independent. The standard deviation deviation for the frustrated case is always larger, even a few orders of magnitude, than the one of the unfrustrated case.

states above the gap produce fast oscillations that average out in the long time limit.

For the frustrated case $N = 2M + 1$ instead, the picture is completely different. The LE exhibits decays and periodic revivals, similarly to its behavior in quenches to critical points [19–28]. Here, because of the closing of the gap, the same perturbation hybridizes several states, which thus contributes to the evolution of the LE. Finite-size effects become important, since by increasing the

chain length the density of states changes and thus also the number of states which get hybridized. These considerations imply a strong sensibility of the LE oscillation frequency and amplitude to all the parameters in the setting.

The results presented in Fig. 1 make it clear that the behaviors of the LE for even and odd $N$ are completely different. To go beyond this qualitative assessment, we can make a quantitative comparison of the difference

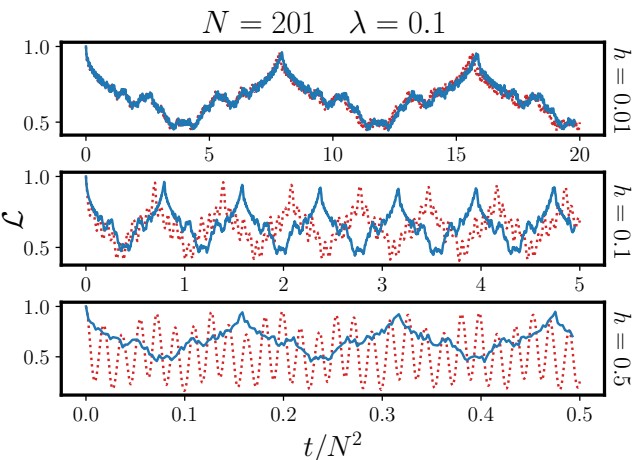

FIG. 3: (Color online) Loschmidt echo's comparison between the numerics (dotted red line) and the analytic expression eq. (5 (blue line) for a spin chain of length $N = 201$ and for $\lambda = 0.1$. The time is rescaled for a better comparison. The results are in agreement for $h = 0.01$, that corresponds to the limit $0 < h \ll \lambda \ll 1$ (upper panel, the curves are mostly superimposed). We also find similar results when $h$ and $\lambda$ are comparable, as shown in the middle panel for the case $\lambda = h = 0.1$. Finally, in the lower panel the failure of the approximation for $h = 0.5$ is shown, where the value of the magnetic field is beyond the assumed range of validity.

between these two behaviors, by considering the time-averaged value of the LE $\bar{\mathcal{L}}$ over a long period, ideally infinite. This analysis, whose results can be found in the left panel of Fig. 2, clearly shows that for the unfrustrated case (blue circles) the time average is almost independent of the size of the ring, while for the frustrated one (red squares) there is a significant dependence on the

ring size, with an asymptotic value in the thermodynamic limit which differs from the even chain length case. The similarity between the frustrated and unfrustrated values for small systems can be easily understood by taking into account that in the frustrated model, for small $N$, the gap between the ground state and the other states in the lowest energy band can be bigger than the perturbation amplitude, hence giving life to an unfrustrated-like behavior for the LE.

As we wrote above, since $H_p$ breaks the spatial invariance, it is impossible to obtain an exact expression for the LE. For the unfrustrated case, it is possible to develop a cumulant expansion [31] which provides the correct evolution of the LE, but its reliability hinges on a clear separation of scales between the strength of the perturbation and the energy gap. When $N$ is odd, the gap closes, and for sufficiently large system size, this approach fails. Nonetheless, to gain some insight into the LE when the system is frustrated, we can resort to a perturbation theory around the classical point $(h = 0)$ [38, 39, 68] and derive an analytic expression that can be compared to our numerical results. Within this approach, we first compute the initial (ground) state of $H_1$ considering, in the beginning, $\lambda \sigma_N^z$ as the perturbation to the Hamiltonian at the classical point $(h = 0)$, and then bringing back the term $h \sum_j \sigma_j^z$ as a second-order perturbation term. By construction, this approach is justified for $0 < h \ll \lambda \ll 1$. The effect of the local term $\lambda \sigma_N^z$ is to split the initial $2N$ degenerate states into three groups. In particular, the ground space becomes two-fold degenerate, separated by an energy gap of order $\lambda$ from $2N - 4$ degenerate states, on top of which, separated by a gap of the same value, there are two other degenerate states. The second perturbation term $h \sum_j \sigma_j^z$ does not act significantly on the two two-dimensional manifolds but removes the macroscopic degeneracy, creating an intermediate band of $2N - 4$ states.

Exploiting this perturbative analysis (see the Method section for details), we obtain for the LE

$$\mathcal{L}(t) = \left| \frac{2}{N(N-1)} \sum_{k=1}^{(N-1)/2} \tan^2 \left[ \frac{(2k-1)\pi}{2(N-1)} \right] \exp \left\{ - \imath 2ht \cos \left[ \frac{(2k-1)\pi}{N-1} \right] \right\} + \frac{2}{N} \exp \left[ \imath t(\lambda + h) \right] \right|^2. \tag{5}$$

In Fig. 3 we compare the analytical results in eq. (5) with the numerical data and we find a substantial agreement between the two in the region $h \ll \lambda$ (see the upper panel). It is also worth noting that the two methods give similar results even when $h$ and $\lambda$ are comparable (middle panel of Fig. 3). The main difference between the two behaviors is, apparently, only a rescaling of the oscillation frequency that seems to be underestimated in the perturbative approach.

In the thermodynamic limit the term proportional to $2/N$ in eq. (5) can be neglected and the expression of the

LE can be approximated as: $\mathcal{L}(t) \simeq \mathcal{F}\left(\frac{2ht}{N^2}\right)$ where the function $\mathcal{F}(x)$ is given by

$$\mathcal{F}(x) = \lim_{M \to \infty} \left| \frac{1}{2M^2} \sum_{k=1}^{M} \tan^2 \left[ \frac{(2k-1)\pi}{4M} \right] \times \right. \tag{6}$$

$$\left. \times \exp \left\{ - \imath x (2M+1)^2 \cos \left[ \frac{(2k-1)\pi}{2M} \right] \right\} \right|^2$$

The function in eq. (6) is somewhat reminiscent of the Weierstrass function [69] and indeed it displays a continuous, but nowhere differentiable behavior. While its

emergence in such a simple context is remarkable, we remark that such fractal curves [70, 71] were already observed in LE evolution [72]. Furthermore, similar curves were also observed in quenches to multicritical points [24, 28], where, as in our case, the LE displays the period of revivals proportional to $N^2$. Presumably, an important reason behind this similarity is that both at studied multicritical points and in the studied topologically frustrated spin chain the spectral gap closes quadratically with the system size.

## 4. METHODS

*a. Loschmidt echo.* Let us provide a detailed description of the method exploited to obtain the data on the Ising model plotted in the paper. Our starting point is to observe that, for spin systems that can be mapped to free-fermionic models, eq. (1) can be rewritten in the following form [21, 22]:

$$\mathcal{L}(t) = |\det(1 - \mathbf{r} + \mathbf{r}e^{-\imath\mathbf{C}t})|. \quad (7)$$

Here

$$\boldsymbol{\Delta}^\dagger = (c_1^\dagger, \ldots, c_N^\dagger, c_1, \ldots, c_N), \quad (8)$$

describes the fermionic operators, $\mathbf{C}$ is the matrix coefficient of the Hamiltonian $H_1$ in the fermionic language, i.e.

$$H_1 = \frac{1}{2}\boldsymbol{\Delta}^\dagger\mathbf{C}\boldsymbol{\Delta}, \quad (9)$$

and $\mathbf{r} = \langle g|\Delta_i^\dagger\Delta_j|g\rangle$ is the two-point fermionic correlation matrix in the initial state. The hermiticity requirement for the Hamiltonian fixes the matrix $\mathbf{C}$ to be of the block-form

$$\mathbf{C} = \begin{pmatrix} \mathbf{S} & \mathbf{T} \\ -\mathbf{T} & -\mathbf{S} \end{pmatrix}, \quad (10)$$

where $\mathbf{S}$ is a symmetric and $\mathbf{T}$ an antisymmetric matrix.

It is useful to rewrite the $\mathbf{r}$ matrix in terms of the correlation functions of the Majorana operators. Following [57] we define:

$$A_i = c_i^\dagger + c_i \quad (11)$$
$$B_i = \imath(c_i^\dagger - c_i). \quad (12)$$

Exploiting Eqs. (11) and (12) and the fact that, since $|g\rangle$ is the ground state of $H_0$, $\langle g|A_iA_j|g\rangle = \langle g|B_iB_j|g\rangle = \delta_{ij}$ it is straightforward to obtain:

$$\mathbf{r} = \frac{1}{4}\begin{pmatrix} 2\mathbf{I} + \mathbf{G} + \mathbf{G}^\mathsf{T} & \mathbf{G} - \mathbf{G}^\mathsf{T} \\ -\mathbf{G} + \mathbf{G}^\mathsf{T} & 2\mathbf{I} - \mathbf{G} - \mathbf{G}^\mathsf{T} \end{pmatrix}. \quad (13)$$

with $G_{ij} = -\imath\langle g|B_iA_j|g\rangle$.

Therefore, to calculate the LE it remains to evaluate the correlation matrix $G$ on the ground state of the unperturbed Hamiltonian $H_0$ and the matrix $C$ linked to

$H_1$. Both can be determined following the same approach. Exploiting the Jordan-Wigner transformation

$$c_j = \left(\bigotimes_{l=1}^{j-1}\sigma_l^z\right)\frac{\sigma_j^x + \imath\sigma_j^y}{2}, \quad c_j^\dagger = \left(\bigotimes_{l=1}^{j-1}\sigma_l^z\right)\frac{\sigma_j^x - \imath\sigma_j^y}{2}, \quad (14)$$

we map the spin system to a quadratic fermionic one. In fact, due to the non-locality of the Jordan-Wigner transformation, the Hamiltonians Eqs. (3) and 4 cannot be written as a quadratic form eq. (9). However, they commute with the parity operator $\Pi^z = \bigotimes_{i=1}^N \sigma_k^z$ and it is possible to separate them into two parity sectors, corresponding to the eigenvalues $\Pi^z = \pm 1$, so that in each sector they are a quadratic fermionic form. In the following, we can restrict ourselves to the Hamiltonians $H_0$ and $H_1$ only in the odd sector ($\Pi^z = -1$) since the ground state of the quantum Ising model eq. (3) with frustrated boundary conditions and $h > 0$ belongs to it [53, 62]. There, they can be written in the form of eq. (9), up to an additive constant. In particular, the matrix $\mathbf{C}$ for $H_1$ in the odd sector, present in eq. (7), can be obtained easily by inspection.

The matrix $\mathbf{G}$ can be found easily from the exact solution of the quantum Ising chain with frustrated boundary conditions [53, 62]. However, for a more efficient numerical implementation, we follow the approach from ref. [22, 57], where we write $H_0$ in the odd sector in the form of eq. (9) and where

$$G_{ij} = -\left(\boldsymbol{\Psi}^\mathsf{T}\boldsymbol{\Phi}\right)_{ij}, \quad (15)$$

with the matrices $\boldsymbol{\Phi}$ and $\boldsymbol{\Psi}$ being formed by the corresponding vectors given by the solution of the following coupled equations:

$$\Phi_k(\mathbf{S} - \mathbf{T}) = \Lambda_k\Psi_k, \quad (16)$$
$$\Psi_k(\mathbf{S} + \mathbf{T}) = \Lambda_k\Phi_k. \quad (17)$$

This problem can be easily solved considering the following eigenvalue problems:

$$\Phi_k(\mathbf{S} - \mathbf{T})(\mathbf{S} + \mathbf{T}) = \Lambda_k^2\Phi_k, \quad (18)$$
$$\Psi_k(\mathbf{S} + \mathbf{T})(\mathbf{S} - \mathbf{T}) = \Lambda_k^2\Psi_k. \quad (19)$$

Here the eigenvalues give us the free-fermionic energies $\Lambda_k$. The sign of particular energy is a matter of choice. Transforming $\Lambda_k$ to $-\Lambda_k$ corresponds simply to switching the creation and the annihilation operator, and to transforming $\Phi_k$ ($\Psi_k$) into $-\Phi_k$ ($-\Psi_k$) in Eqs. (16) and (17). It is important to note that the parity requirements do not allow for the ground state of $H_0$ to be the vacuum state for free fermions with positive energy [53, 62]. Thus, assuming the eigenvalues of the matrix appearing on the l.h.s. of Eqs. (18) and (19) are labeled in ascending order, the ground state corresponds to the vacuum state of fermions with $\Lambda_1 < 0$ and the remaining energies $\Lambda_k$ positive.

*b. Perturbation theory near the classical point* Let us now turn to provide some more details on the perturbative approach to the LE near the classical point in the presence of topological frustration. The first step consists of finding the ground state of the Hamiltonian $H_0$ in eq. (3), treating the term $h \sum_j \sigma_j^z$ as a perturbation. It is known that, at the classical point, in the presence of an odd number of spins the interplay between periodic boundary conditions and antiferromagnetic interactions gives rise to a $2N$-fold degenerate ground-state manifold. Such a space is spanned by the kink states $|j\rangle$ and $\Pi^z |j\rangle$, $j = 1, 2, \ldots N$ with energy $-(N-2)$, that have one ferromagnetic bond $\sigma_j^x = \sigma_{j+1}^x = \pm 1$ respectively, the others being antiferromagnetic ($\sigma_k^x = -\sigma_{k+1}^x$ for $k \neq j$). The excited states outside this manifold are separated from the ground space by an energy gap of order unity so that we can neglect them in a perturbative approach. By considering the magnetic field the $2N$-fold degeneracy is removed and a narrow band of states is created, with a gap that separates the ground state from the other elements of the band closing as $1/N^2$ (see Ref. [37, 38]). To the lowest order in perturbation theory in $h$ we found for the initial state appearing in eq. (1), that is for the ground state of the unperturbed system, the expression:

$$|g\rangle = \frac{1}{\sqrt{N}} \sum_{j=1}^{N} \frac{1 - \Pi^z}{\sqrt{2}} |j\rangle. \quad (20)$$

The next step is to find the lowest energy states of the Hamiltonian $H_1$ in eq. (4) through a perturbation theory both for $h > 0$ and $\lambda > 0$. Since we first apply the perturbation theory in $\lambda$ while we consider $h$ as a second-order perturbation, we are assuming that $h \ll \lambda \ll 1$. Also in this case we start again from the $2N$ degenerate ground space formed by the kink states and we treat the term $\lambda \sigma_N^z$ as a perturbation. Again we find that the degeneracy is removed and, at this point, the system shows two-fold degenerate ground states:

$$|\psi_\pm\rangle = \frac{1 \pm \Pi^z}{2} (|N-1\rangle \mp |N\rangle), \quad (21)$$

separated by an energy gap equal to $\lambda$ from $2N-4$ degenerate kink states. Above this macroscopically degenerate manifold, separated by a gap $\lambda$ there are other two states:

$$|\phi_\pm\rangle = \frac{1 \pm \Pi^z}{2} (|N-1\rangle \pm |N\rangle) \quad (22)$$

We now consider the second-order perturbation $h \sum_j \sigma_j^z$. Its effect on the $|\psi_\pm\rangle$ and $|\phi_\pm\rangle$ states is only a shift in the energy respectively of $\mp h$. Furthermore, it creates a band of states from the kink ones given by:

$$|\xi_\pm, m\rangle = \frac{1 \pm \Pi^z}{\sqrt{N-1}} \sum_{j=1}^{N-2} (-1)^j \sin\left(\frac{m\pi}{N-1} j\right) |j\rangle, \quad (23)$$

with $m = 1, 2, \ldots, N-2$. The energies of the discussed eigenstates are given by

$$E(\psi_\pm) = -(N-2) - (\lambda + h), \quad (24)$$

$$E(\phi_\pm) = -(N-2) + \lambda + h, \quad (25)$$

$$E(\xi_\pm, m) = -(N-2) \mp 2h \cos\left(\frac{m\pi}{N-1}\right). \quad (26)$$

The calculation of the Loschmidt echo is now straightforward. From the definition eq. (1), expressing the initial state eq. (20) in terms of the eigenstates of the perturbed model Eqs. (21), (22), and (23) and applying the evolution operator $e^{-\imath H_1 t}$ we finally obtain the expression in Eq. (5).

## 5. CONCLUSION

We have analyzed the behavior of the LE in short-range antiferromagnetic one-dimensional spin systems with periodic boundary conditions in the presence of a perturbation that violates translational invariance but leaves unaffected the parity, namely a local magnetic field. Under these conditions, the LE shows an anomalous dependence on the number of elements in the system. When this number is even, LE shows small random oscillations around a value very close to unity that is almost independent of the system size, and the amplitude of these oscillations tends to decrease as the system grows until it disappears in the thermodynamic limit. On the contrary, in the presence of a ring made out of an odd number of sites, the oscillations are large and do not disappear in the thermodynamic limit while the average value is strongly dependent on the system size. The presence of two different behaviors can be traced back to the different energy spectra for even and odd $N$. Namely, for an odd number of elements topological frustration is induced and it leads to a closure of the energy gap, which is instead finite in these systems when $N$ is even. These general results have been tested in a paradigmatic model, the Ising model in the transverse field, using both exact diagonalization methods and perturbation theory.

The LE can thus be used to distinguish the spin chains with $N$ and $N+1$ sites, for arbitrarily large $N$. While this is not the first time the even/oddness in the number of elements of a system becomes important [34], our result is especially relevant taking into account that LE is an experimentally accessible quantity, by looking at the decoherence of a two-level system interacting with the spin system [17, 19, 22, 30]. Moreover, since the LE plays a fundamental role in several problems of current interest in quantum thermodynamics such as quantum work statistics [31, 32] and information scrambling [32, 73], its different behavior signals a different response of systems with or without topological frustration in various quantum energetics protocols. In particular, the larger fluctuations of the LE with frustration indicate that these systems can outperform their non-frustrated equivalent. Indeed, a detailed analysis of the implication of this work

in these applications and additional quantitative characterizations of the frustrated LE will be the subject of future works.

On the other hand, this particular LE pattern can be seen as a further interesting property of one-dimensional systems with topological frustration. Despite their simplicity, they present several peculiar aspects such as incommensurate magnetic patterns [38, 68], the appearance of phase transitions not present with boundary conditions that do not force the presence of frustration [38, 39], etc. Until now, the analysis had focused on the static aspects induced by topological frustration. However, our work emphasizes that the gapless nature of such systems can greatly influence their dynamics, which can possibly open the door for their applications in quantum computing [33] as well as in quantum thermodynamics.

## ACKNOWLEDGMENTS

S.M.G., F.F., and G.T. acknowledge support from the QuantiXLie Center of Excellence, a project co-financed by the Croatian Government and European Union through the European Regional Development Fund – the Competitiveness and Cohesion (Grant No. KK.01.1.1.01.0004). V.M., D.K., S.M.G., and F.F. also acknowledge support from the Croatian Science Foundation (HrZZ) Projects No. IP–2016–6–3347. S.M.G. and F.F. also acknowledge support from the Croatian Science Foundation (HrZZ) Project No. IP–2019–4–3321.

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
