# Peer review of "An odd thermodynamic limit for the Loschmidt echo"

_SciPost Physics_

## Round 2 · Referee Report · Anonymous · 2021-10-26

Report

I am satisfied with the changes made in response to the reports. The only further comment I have is that the work by

Heyl et al. Physical review letters 110 (13), 135704,

related theoretical work, as well as the experiment

Jurcevic et al. Physical Review Letters 119 (8), 080501

should also be mentioned because those two papers and related theoretical works also deal with the Loschmidt echo after a quantum quench and some of them also consider the transverse Ising model.

As I wrote in my previous report, I believe that this works provides a new connection between topological frustration and quantum dynamics and is therefore of interest for the condensed matter/statistical physics community. I recommend publication in SciPost Physics.

---

## Round 2 · Referee Report · Anonymous · 2021-11-10

Report

The authors have appropriately addressed all the issues. I therefore
recommend the present version of the manuscript for publication.

---

## Round 2 · Author Response

We thank the referees and the editor for their work. We appreciate that both referees acknowledge the soundness of our results and we admit that we were probably too carried away by our excitement in the original writing of the manuscript. We felt that the fact that we finally found a way to experimentally detect the change in the low energy spectral properties of frustrated chains and in this way to distinguish the parity in the chain length was intrinsically cool, and thus we placed admittedly too much emphasis on this aspect and forgot to highlight other reasons of interest.
In this revision we accepted the criticisms from the two reviewers and completely rewrote the introduction and substantially reworked the abstract and conclusion. We strove to better place our work into the vast activity in quantum out-of-equilibrium systems and to keep our excitement in check. We also stressed better the implications of our results (mostly in relation to quantum thermodynamics and energetics), and that, with respect to the dependence of the Loschmidt Echo on the parity of the chain length, our findings are generic for systems featuring only topological frustration (that is, 1D chain in which frustration only arise because of the boundary conditions and in which thus the amount of frustration is not extensive). However, since the LE reflect mostly the different low energy spectra with and without frustration, the two behaviors we feature are qualitatively expect to emerge either in standard gapped systems, or in systems were frustration produces a gapless excitations (as it happens in certain regions of the phase diagram of the ANNNI model, mentioned by one of the referees).
We did not significantly modify the main body of our calculation, but we divided it into sections to help in clarifying the flow of our reasoning and to better separate general considerations from specific examples.
In conclusion, we think we successfully addressed the referee's criticisms and we hope that our paper will now be accepted.

---

## Round 2 · List of Changes

We substantially rewrote the paper, completely changing the introduction, but did not introduce new calculations or figure.

---

## Editorial Decision

accepted_in_other_journal;_awaiting_puboffer_acceptance